# Yoga and Multiple Sclerosis: Maintaining engagement in physical activity

**Jenni Naisby**[ID]**[1]** *****, **Gemma Wilson-Menzfeld[2], Katherine Baker[1], Rosie Morris[1], Jonathan Robinson**[ID]**[3], Gill Barry[1]**

**1** Department of Sport, Exercise & Rehabilitation, Northumbria University, Newcastle upon Tyne, United Kingdom, **2** Department of Nursing, Midwifery and Health, Northumbria University, Newcastle upon Tyne, United Kingdom, **3** School of Health and Social Care, Teesside University, Middlesbrough, United Kingdom

* Jenni.naisby@northumbria.ac.uk

## Abstract

### Background

Physical activity is encouraged for people with Multiple Sclerosis. Yoga is a popular form of physical activity and is chosen by some people with Multiple Sclerosis. However, little is known about the impact of yoga for this population, alongside what influences ongoing engagement.

### Aim

The aim of this study is to qualitatively explore the impact of online home-based yoga on people with Multiple Sclerosis and to explore factors that influence engagement.

### Methods

A qualitative study using semi-structured interviews and focus groups with people with Multiple Sclerosis and a yoga teacher. Thematic analysis was used to analyse the data. Ethical Approval was gained from Northumbria University.

### Findings

Three overarching themes emerged from the analysis. 'Yoga as engagement in physical activity' captured the reasoning for participating in yoga and how this method of physical activity was an alternative to physical activity done prior to diagnosis. Frustration was apparent within this theme that some individuals were unable to engage in the range of physical activity that they wished to. 'Yoga is a personalised approach' demonstrated the flexibility and inclusivity of yoga, for individuals with varying symptoms to be able to engage with. Finally, 'yoga impacts individuals both physically and psychologically' captured the focus on the psychological impact of yoga, improving wellbeing and control.

### Conclusions

Yoga gives people with Multiple Sclerosis the feeling of control over their symptoms and a means to engage with meaningful physical activity. Prior involvement in physical activity

**Data Availability Statement:** Relevant excerpts could be made available upon request but only those that have been anonymised. The ethical approval process was dependent upon no raw data being shared outside of the research team and we

have not received ethical approval to store excerpts in a data repository. This study was approved by Northumbria University's ethical approval system. Data requests can be sent to dp. officer@northumbria.ac.uk.

**Funding:** The author(s) received no specific funding for this work.

**Competing interests:** The authors have declared that no competing interests exist.

influenced engagement in yoga and wanting to push themselves. There was reluctance among this group to engage with aerobic activity, which warrants future investigation and support from health and exercise professionals.

## Introduction

Multiple Sclerosis (MS) is an autoimmune disease of the central nervous system, which is progressive in nature [1]. It is estimated that 2.8 million people worldwide live with MS [2]. The symptoms of MS can vary, affecting both physical and mental health [3] with symptoms including fatigue, movement problems [4], pain [5] reduced strength and balance [6] and reduced quality of life [7]. Whilst physical activity is encouraged for people with MS (pwMS), some have demonstrated fear and apprehension regarding engagement in physical activity, alongside concerns about fatigue [8].

PwMS who engaged with physical activity report improvement in strength, fatigue, quality of life, gait and balance [9]. The Chief Medical Officer Physical Activity Guidelines advocate a range of physical activity, including yoga, for individuals living with long term conditions [10]. Yoga has been shown to improve both physical and mental health [11]. Whilst some studies have demonstrated the positive effect of home yoga on strength and balance in MS [12], the evidence regarding the effectiveness of yoga and MS is scarce [13]. However, as yoga is increasing in popularity with this population [13] there is a need to explore the motivators and barriers to yoga in pwMS.

Views on engagement with yoga specifically have had limited exploration for pwMS [8]. It is advocated to address the individual needs of pwMS and to develop their knowledge of physical activity [8, 14]. There is no minimum amount of physical activity stated to achieve some health benefits, and for those with long term conditions, such as MS, engagement in small amounts of physical activity and building on this is encouraged [10]. It has been suggested home-based exercise is beneficial for pwMS, as it is safe and feasible alongside promoting access to exercise [15]. However, qualitative exploration of home-based yoga in MS is limited. The impact of yoga is complex, and personal experiences of the yoga process in neurological conditions including MS is required [16]. A hybrid approach to the delivery of yoga has demonstrated promise [17]. Understanding pwMS individual needs is of upmost importance in relation to engagement in physical activity [8, 14], in this case yoga. Therefore, the aim of this study is to explore the impact of home-based yoga on pwMS and to understand engagement with this form of physical activity.

## Methods

### Study design

A qualitative, Phenomenological methodology [18] design was used to explore the role of yoga and engagement in this form of physical activity for pwMS. Ethical approval was received by Northumbria University's ethical approval system (Ref: 29257).

### Participants

Eight participants took part in this study (1 instructor; 7 class participants). All participants were over 18 years old, five were female and two were male. All class participants were members of a yoga studio based in London and were living with MS. All participants had taken part

in face-to-face yoga prior to COVID-19 and then participated in this online at home. All participants had carried out yoga for at least two years. The instructor sent potential participants an information sheet and consent form via email, explaining the purpose of the study. Participants were offered the opportunity to ask questions before providing written consent.

### Yoga class content

The yoga style is Iyengar Yoga. The teacher aims to include all with yoga and adapt as needed with props. Poses include arm stretches, supine leg extension, hamstring stretch supine. Standing poses include triangle pose, Trikonasana and Warrior poses with the help of a wall or table behind for balance and sometimes a chair in front with the backrest facing the student. These movements can be adapted to supine. Sitting poses include legs straight, bent, crossed and backwards Virasana. Inversions include shoulder stand with a chair, keeping the weight of the body on the chair or legs on the wall or bent on a chair.

### Data collection

Participants were invited to attend either a focus group or semi-structured interview. The focus group included class participants only, and two separate one-to-one semi-structured interviews were setup for class participants who were not available at this time. The instructor also participated separately in a one-to-one interview. The focus group and all interviews were carried out using Zoom Video Communications, Inc. Interviews were conducted by GWM (Health Psychologist) and GB (Biomechanist).

A topic guide was informed by a review of the literature and covered motivations to begin yoga, the impact of yoga on MS symptoms, online yoga provision, in-person yoga provision, social connections, and future delivery (Fig 1). A paper reporting the role of online yoga provision from this interview schedule has been published elsewhere [17]. Sessions lasted between 27–53 minutes. The Zoom recordings were transcribed verbatim and deleted once the transcript was produced.

### Data analysis

Transcripts were uploaded into NVivo 12 (QSR International) for analysis. Braun and Clarke's reflexive, inductive Thematic Analysis was chosen as the data analysis strategy due to its theoretical freedom [19, 20]. JN analysed all transcripts. Initially, the analyst immersed themselves within the transcripts before generating initial codes and subsequent themes. The potential influence of the researcher on this process, was addressed by frequent qualitative debriefing with the wider research team who have health and exercise backgrounds.

### Results

Three themes were developed from the interviews and focus group (Table 1). Several sub-themes contributed to each theme.

**1. Yoga as engagement in physical activity**. Yoga allowed individuals to engage in physical activity in a manageable way. Some participants felt unable to participate in many forms of physical activity, which was both disheartening and frustrating. Discussion of previous physical activity, whilst providing some understanding and drive for participation yoga, could be distressing for some individuals due to the feeling of not being able to be involved any longer. Yoga was considered as providing some of the participants with a workout of strength and balance as well as offering some control and choice to participate in physical activity. Participants wanted a sense of working hard and achievement from taking part in yoga.

**Focus Group / semi-structured interview schedule**

Confirmation of consent and confidentiality of the information. General introductions and introduce the purpose of the Focus Groups.

1. What were your motivations to begin yoga initially?
2. Do you feel that yoga has an impact (positive or negative) on any MS symptoms?
   a. Fatigue
   b. Balance
   c. Other?
3. Do these sessions differ when delivered online?
   a. Do you prefer the online classes or face-to-face classes, or not? Why, why not?
   b. What do you prefer in person?
   c. What do you prefer online?
4. Were you able to easily access this online?
   a. What tech are you using? Has it been easy to use?
   b. Did you have to buy any additional equipment/broadband?
   c. Visual symptoms make it difficult?
5. Was it as easy to follow the poses/flow online as it was in person?
6. Is it important that this is live, with the instructor 'in the room' or would you like to be able to access pre-recorded sessions?
7. Do you think your experiences of online yoga was influenced by having previous face-to-face experience?
8. Is there a cost difference between online and face-to-face classes? Does this make a difference?
9. What about the social aspect? Is this different online?
10. To discuss any involvement in similar services (both face-to-face and online) and describe experiences of these services.
11. To discuss the interviewee's perceived impact of the sessions (on physical activity, well-being, social isolation, loneliness, or other).
    a. Face to face sessions.
    b. Online sessions.
    c. Have this changed since being online?
12. To discuss recommendations for ongoing service provision, including anything they would like to stay the same, or anything that they would like to change.
    a. Would you go back face-to-face or stay online?
    b. Why?

Any further questions. Thank everyone for participating in this focus group.

**Fig 1. Focus group / semi-structured interview schedule.**

*An alternative form of physical activity*. Engagement in physical activity was important to individuals. Being able to access physical activity and feeling as though people had the capability to fully partake was necessary. Accessing different forms of physical activity could be challenging at times and yoga provided a means to participate. Two participants felt that yoga provided a comparable work out to their practices in the gym.

**Table 1. Development of overarching themes.**

| Yoga as engagement in physical activity | An alternative form of physical activity |
|---|---|
| | Previous engagement and frustration with physical activity |
| Yoga is a personalised approach (for a diverse condition) | Understanding Multiple Sclerosis |
| | Variation of the condition |
| | Providing alternatives in yoga |
| Yoga impacts individuals both physically and psychologically | Impact on mobility |
| | 'Feel better' |

*We go for a walk in the park, it's a long way to get there and we don't have a car. This business of having a live person, whos really good at saying come on, stick with it.*

*(P2)*

*But finding the yoga actually and I have found that some of the poses, they are just like working out in the gym.*

*(P3)*

*It is possible actually to get the results without being so tired and because of the weakness of my legs I am not able to do it, but there is a way and so good poses*

*(P3)*

However, it was noted that yoga was not felt to support aerobic fitness. Individuals did discuss struggling to engage with this type of exercise. This was viewed as a limitation to peoples' overall fitness, and some were unsure of how to participate in this type of physical activity.

*So I consider myself to be really unfit, you know, so yoga does a lot of other things, but I am not sure how good it is for cardio.*

*(FGF1)*

*So I don't think I will be able to do it, the cardio one*

*(P3)*

*Previous engagement and frustration with physical activity*. Participation in physical activity in the past was difficult for some individuals to discuss due to them feeling unable to engage in activities they would enjoy such as the gym or cycling. The strength to do these activities and the fatigue following had influenced some participants decision to not partake in certain types of exercise. This drive to be involved with physical activity spurred some participants to put everything into their yoga class and feel as though they were getting the best out of this. The participants with a history of attending the gym, discussed this aspect in the most detail.

(discussing the gym) *Really, and I love it. I love working out and now it just kills me so much I can't do it.*

*(P3)*

(discussing a bike) *I used to do it, but I don't have the strength. . .I get more frustrated because I see that I can't do it. . .*

*(P3)*

*Depending on, I mean, you know, with MS particularly you can have a really good time, I don't know about the other conditions, but you know you can have amazing energy and then you can wake up one morning and just have no energy at all. So the whole, for me doing anything like even going to a gym to get on a bicycle, you know, I am already factoring in the energy that it takes to get there. The energy it takes, because sometimes just putting your gym kit on takes a load of energy that one doesn't have.*

*(FGF1)*

One individual discussed in detail their previous physical activity and their understanding of the positive role of exercise and physical activity on the neurophysiology of MS. This contributed to their desire to frequently practice yoga.

*Yeah, me as well, because for me it was important to work out, that's how I'm used to and you know, my brother because he said, you have to keep moving he said, because you know the signal, even if it is lost all along the way because the nerves are damaged, it will look a way to and it will create new routes, but you need to you know exercise.*

*(P3)*

**2. Yoga is a personalised approach (for a diverse condition)**. Personalisation of yoga to individuals was key for successful engagement. This involved the teacher understanding the condition and being able to adapt poses and provide alternatives for individuals. Trust in the teacher and their ability was apparent within the interviews and the importance of this rapport that has been developed.

*Variation of the condition*. The variation of MS and how no two individuals experience the same symptoms was captured. Personalising the classes and ensuring an individualised approach was important to participants, as some of the poses and classes could require things that some people didn't feel they were able to do. Alongside this, how symptoms changed and presented each day could vary which needs to be taken into account when engaging in a yoga class.

*I'm not that mobile. Some of the people with MS they are more mobile and everyone here has a different, you know, difficulty, but for me it's the left part, the left arm, which if they ask me to put my hand up, you see this one, to hold it straight is really kind of difficult and my posture as well. I have to be near a wall or something and when [anonymised] or some of the teachers are there*

*(P3)*

*Understanding of Multiple Sclerosis*. The teacher and the class understanding MS was deemed of upmost importance to a number of individuals. It was important that a teacher understood the condition to personalise the yoga poses and session. Having someone who understood the nature of the condition, the variation and symptoms was also deemed key, as this would influence how much individuals felt they could be 'pushed'.

*So I do have to push them sometimes, but sometimes if they are really too tired, like they are sick or something, then obviously I am not going to push*

*(T1)*

*so working with someone who you, who helps you to understand the nature of your limitations or where you can improve and so on and so forth.*

*(FGF1)*

*but because I'm so slow, so it doesn't really work, even [anonymised] told me, I can do it, but there will be more, faster and I will more frustrated with. . . I am actually, that is why when she is teaching she knows me. She says take your time, whatever, not only me but there are other people who are not that, as quick as, you know each other.*

*(P2)*

*She spends a bit of extra time with them. Really focusing on understanding their condition, what they can do. It's like they get, during those first few classes, a lot of extra attention from her so she can work out, what they can do, what works for them, what's best for them.*

*(FGM2)*

*but it is very useful I find to, regardless of whether it's online or in the physical space, to be around people that sort of understand what you're struggling with.*

*(FGF1)*

*you can really tell and [anonymised] advocates that as well, that you work to the individual's need rather than the Iyengar set of rigid instructions*

*(FGF2)*

*Really focusing on understanding their condition, what they can do. It's like they get, during those first few classes, a lot of extra attention from her so she can work out, what they can do, what works for them, what's best for them.*

*(FGM2)*

One individual spent some time discussing problems with remembering poses. Due to their previous engagement in yoga, they were able to work through the classes; however, highlighted this challenge individuals living with MS may face.

*I've done yoga for a long time so I'm remembering what I was taught before my brain got demyelinated. I don't have to learn it again as long as I am very careful, I stick with what I remember and make myself do what I know to do to stay safe.*

*(P2)*

*I've got to remember what he's saying. . . I said I yearn for a routine I can follow and do the same again and again*

*(P2)*

*Providing alternatives in yoga.* Yoga was viewed as being accessible through providing alternatives such as using a chair, equipment and adapting the postures. Yoga could be viewed as requiring as standardised sequence of postures; however, both the teacher and some participants commented on the flexibility and individualised nature of these sessions and the importance of this.

*You know, she's varying the poses for different people with different issues and that inspires confidence, the fact that you know she understands what is going on for you and she can talk you through different aspects of that and some do it slightly differently*

*(FGM2)*

*we do quite a lot of sitting things. You watched it. . . like arms stretches, twists where you can either sit on the floor or on the chair, wherever you are and then a resting pose, some sort of resting pose*

*(T1)*

*Unless you know one day they have a problem and then we'll do an alternative. There are always alternatives. I mean, that's the tiring thing, you don't just say, you do this, we will all do it, everybody will do it a little bit differently, but as long as they do it and I'm, okay there is another point, we are very exact in Iyengar yoga. It doesn't matter so much how they do it, whether it is a pillow or a bolster or this or that, as long as the shape is there and they get something from it.*

*(T1)*

*But I think, you know, [anonymised] specialises, you know, is especially interested in the individuals. So she aims everything that she does to compensate or to help whatever individual need a person has or [break in audio]. Everything is adaptable but I am not sure all, all teachers and even Iyengar teachers are that compassionate and have the experience that [anonymised] has in order to. . . Yeah, that and I see [anonymised] as well*

*(FGF2)*

**3. Yoga impacts individuals both physically and psychologically**. Engagement in yoga and the impact on individuals psychologically was discussed in more depth and more frequently than the physical impact. Whilst there were noted physical benefits, there was less detail and focus centred around this area.

*'Feel better'*. The impact of yoga on some individuals' wellbeing was highlighted. Feeling better was a term frequently used to describe the effect of yoga and reasons for continued engagement. There was a sense of promoting yoga and wanting other individuals to be aware of the benefits participants described, in particular, this psychological impact.

*You feel better than you've ever felt. . . It makes you feel better than anything, it's amazing*

*(P2)*

*I just push myself and I go and after I feel really good, you know, mentally, physically*

*(P3)*

*And I have been able to do it before and with yoga when starting, I feel so good*

*(P3)*

*Yeah, as a. . . when you do yoga, you always feel better*

*(T1)*

Alongside this, relaxation and breathing involved in yoga were viewed as an important element to some of the individuals. This was a technique individuals could practice at other times at home and was described as contributing to increased energy. Within the interview, the impact of fatigue and this limiting engagement in activities was highlighted, and yoga was a means to take some control of this symptom.

*This business of relaxation is not be taken seriously*

*(P2)*

*But this one is for a Nyama breathing as well. It is a good one. So I can do it myself when I am, you know, at home.*

*(P3)*

*But definitely the breathing is very, very important and I will always finish them at least 15 minutes early and get them to do some breathing and because they need this at the end, because there is exhaustion, there is, yeah. . . they need that to recharge.*

*(T1)*

*I am full of energy after that.*

*(P3)*

*Impact on mobility.* Regular yoga practice was discussed by some of the participants as helping with their mobility including walking, joint range of movement and spasticity. This was not something that was discussed at great length within the interviews or focus groups.

*The yoga for MS is really good at stretching out those tendencies and why that helps someone with MS so much is because those are the muscles that if you stretch them your walking works better*

*(P2)*

(discussing walking) *I lifted my legs better and how to say, after doing 1½ hours some how your body, everything is just going*

*(P3)*

*Lie on your back and you use a belt and what it does is it pulls back hamstring, that stops you getting spasticity*

*(P2)*

## Discussion

This study aimed to explore the impact of home-based yoga in pwMS and to understand engagement with this form of physical activity. Whilst there is a focus on the role of yoga supporting improvements in strength and balance [12] and other physical symptoms, there was a limited discussion of these within the interviews. The key focus was the psychological benefits pwMS found with home-based online yoga. Key for successful engagement in physical activity for pwMS is self-efficacy [21]. A recent mixed methods study for yoga with pwMS found increased self-efficacy for physical activity, in turn increasing confidence in engagement with physical activity [22]. An improved sense of wellbeing prevailed within the interviews and having a sense of control over symptoms such as fatigue, which can be a barrier to engagement in physical activity [8, 23]. PwMS were able to engage with yoga with confidence, due to having this sense of control over symptoms and the ability to do this made individuals feel better and in turn continue their engagement. PwMS could use the techniques from the home-based yoga at times other than classes and found the relaxation element of yoga beneficial for wellbeing. Alongside this, an improved self-efficacy and control could be due to individualisation of the yoga, tailored to individuals' symptoms. Yoga is inherently individualised, and instructors have demonstrated knowledge of conditions such as MS [24]. The real-world application of yoga as discussed in the interviews, should be considered for future large scale trails, and ensuring this individualised nature prevails [24]. Within the interviews the knowledge of the instructor and adapting postures was highlighted as key for supporting engagement. This is an important implication for future research, consideration of the individual.

In the context of control, the use of breathing exercises was discussed by some participants. Within yoga for traumatic brain injury these breathing exercises have been found to manage stress and support individuals emotionally [25]. Alongside this, within our current study breathing was mentioned in relation to fatigue management. A pilot trail of yoga including breathing exercises for pwMS found breathing exercises improve fatigue and to influence overall wellbeing [26]. It is postulated that combined exercises, including respiratory and the upper limb may influence fatigue due to the influence of MS on respiratory dysfunction, and breathing alongside upper limb exercise as a rehabilitation strategy may influence fatigue associated with this [27]. Yoga provides a means to intrgrate these and engage in both activities.

Engagement in some form of physical activity was important to the pwMS interviewed. Whilst yoga allowed this sense of participation and control, there was frustration evident about past physical activity. Lives before and after MS have been explored in a qualitative study with pwMS [28] finding uncontrollable changes in pwMS lives and a feeling of missing out on adulthood. The current study develops these findings by considering previous physical activity engagement, as there was certainly a frustration at being unable to engage with physical activity that individuals had done in the past. PwMS identity were that of a physically active person, and this was difficult to accept having to change the type of physical activity. Identity has been shown to be important for sustaining long term engagement with physical activity [29]. This is important to consider for supporting pwMS as it was apparent those interviewed had found their identity as engaging in yoga, however for others this may be more challenging and there is the need to understand individual preferences.

Due to previous engagement in physical activity, it was important to the pwMS that the yoga felt challenging and like a workout with this giving a sense of achievement and working hard. A randomised controlled trail of yoga versus physical therapy, which included a range of strength and conditioning exercises found no difference between these groups on the influence on quality of life [30]. However, self-belief is important for sustaining physical activity behaviours in the longer term, including confidence in ability to engage with physical activity [29]. Seeing progress and feeling a physical response to the yoga provided pwMS with satisfaction and encouragement that they were able to engage with home-based yoga to a level that suited their needs. There was some despondence in the group regarding aerobic exercise. Whilst yoga was felt to support strength, aerobic exercise was felt to be missing and a challenge to engage with for those interviewed. This highlights the need for support for pwMS to enhance confidence with this type of physical activity [31] alongside yoga, as aerobic activity is a key feature of physical activity guidelines for long term conditions [10] and specifically for MS [32]. Aerobic exercise in pwMS has been shown to improve physical, mental and social functioning [13]. However, this study has identified that some pwMS, who have a past history of physical activity, and are currently engaged with yoga, have concerns about aerobic exercise. Supervised aerobic exercise programmes have shown promise in pwMS [33]. Given the value of individualised approaches and support from the home-based yoga from this interview, these elements could be integrated into aerobic interventions, as some pwMS may require guidance and support in this area, irrespective of previous engagement.

The unpredictability of MS has been found to be a challenge for pwMS [34]. With physical activity and MS, barriers can mean this feels to be not possible [35]. However, awareness of symptom variation and individual needs can support tailoring of physical activity. Within the interviews, pwMS discussed the variation in the condition both between one another and themselves day to day. The individualised nature of yoga and the ability to adapt the postures allowed pwMS with a range of symptoms to participate. This finding has also been echoed in the context of relapse to support the feeling of accomplishment and making physical activity accessible for the individual being key for engagement [31]. This individualised approach

worked well with the flexibility of home-based yoga, feeling understood and safe by the teacher and peers and that something was being achieved for the individual.

## Strengths and limitations

To date, there is limited qualitative exploration of the impact and engagement in yoga for pwMS. This study has provided valuable insights into what pwMS value for this type of physical activity alongside engagement being influenced by the psychological benefits felt and perceived the need to engage with physical activity. The trustworthiness of the study is enhanced through the different perspectives of the research team from different backgrounds of health and exercise influencing the qualitative analysis. Thematic analysis allows a clear audit trail of the research enhancing credibility and dependability.

This study only included pwMS who were actively involved in a home-based yoga group. Therefore, views of those who have chosen to not continue or engage with this type of exercise are not considered. Future research should aim to explore this group to further develop understanding and development of yoga for pwMS. The time of this study was during the COVID-19 pandemic, so the context of this specific time should be considered. Alongside this, the views of pwMS were from one class, with one teacher. To enhance transferability a national exploration of yoga for MS should be considered.

## Conclusions

Engagement in yoga by pwMS affords individuals some control over symptoms, including fatigue, which has been demonstrated to be a barrier to engagement in physical activity for pwMS. PwMS who were physically active prior to diagnosis want to engage with meaningful physical activity, which includes feeling a sense of working hard, which links with their previous engagement with physical activity. This is an important consideration for the design of future yoga interventions for pwMS. A number of pwMS wanted to feel challenged by the physical activity. Yoga allowed pwMS to tailor the exercise to their specific needs, alongside having flexibility to engage to allow for changes in symptoms.

## Acknowledgments

The authors would like to thank Iyengar Yoga London.

## Author Contributions

**Conceptualization:** Jenni Naisby, Gemma Wilson-Menzfeld, Katherine Baker, Rosie Morris, Jonathan Robinson, Gill Barry.

**Formal analysis:** Jenni Naisby.

**Investigation:** Gemma Wilson-Menzfeld, Gill Barry.

**Methodology:** Gemma Wilson-Menzfeld, Gill Barry.

**Project administration:** Gemma Wilson-Menzfeld, Gill Barry.

**Writing – original draft:** Jenni Naisby, Gemma Wilson-Menzfeld, Gill Barry.

**Writing – review & editing:** Jenni Naisby, Gemma Wilson-Menzfeld, Katherine Baker, Rosie Morris, Jonathan Robinson, Gill Barry.

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
