## [Decision Letter · Decision Letter 0]

2 Feb 2023

PONE-D-22-34381Yoga and Multiple Sclerosis: Maintaining Engagement in Physical ActivityPLOS ONE

Dear Dr. Renni Nasiby,

Thank you for submitting your manuscript to PLOS ONE. After careful consideration, we feel that it has merit but does not fully meet PLOS ONE’s publication criteria as it currently stands. Therefore, we invite you to submit a revised version of the manuscript that addresses the points raised during the review process.

           Reviewer 1:

<ul> <li> 

I believe that this is an interesting study and it has a merit for this journal. However, this manuscripts needs substantial development and editing. Especially, a strong introduction, discussion and conclusion parts needed with up to date literature review and (2020 and above) and practical suggestions for the future researchers. A strong theoretical background needed in manuscript. In addition, reliability and validity of the data should be reported. In summary, the study needs more clarification in terms of theory, background and reliability and validity of the data. Also, each individual characteristics should be defined in detail such as gender, race, education and culture. 

Reviewer 2:

 <li> 

The original study on the effect of Yoga on Multiple Sclerosis is well conceptualized. The study aimed to explore the impact of online home-based yoga on people with Multiple Sclorasis qualitatively.

 I recommend the authors to give more details on the protocol and Yoga sessions conducted. The frequency, the yoga poses adapted or not adapted to the needs of the participants.

Since, YOGA can be individualized as movement intervention, I would like to see the how the increased awareness and knowledge on MS could contribute to the further individualization to YOGA sessions.

It is possible to discuss the role of the breathing exercises more on the positive feelings and wellbeing. Moreover, if the combination of Yoga poses and breathing augmented the effect, it would be good to have some science based ideas about this link.

Please describe if there is any specific information on the degree or level of participant's MS even the average age.

The words "sense of working hard" and "sense of achievement" could be mentioned in the results and discussion to strengthen the link between the conclusion and the results.

Data is available due to the condition for the ethical approval.

We look forward to receiving your revised manuscript.

Kind regards,

Bijen Filiz

Academic Editor

PLOS ONE

Journal Requirements:

Reviewers' comments:

Reviewer's Responses to Questions

**Comments to the Author**

1. Is the manuscript technically sound, and do the data support the conclusions?

Reviewer #1: Partly

Reviewer #2: Yes

2. Has the statistical analysis been performed appropriately and rigorously? 

Reviewer #1: N/A

Reviewer #2: N/A

3. Have the authors made all data underlying the findings in their manuscript fully available?

Reviewer #1: No

Reviewer #2: No

4. Is the manuscript presented in an intelligible fashion and written in standard English?

Reviewer #1: Yes

Reviewer #2: Yes

5. Review Comments to the Author

Reviewer #1: The original study on the effect of Yoga on Multiple Sclerosis is well conceptualized. The study aimed to explore the impact of online home-based yoga on people with Multiple Sclorasis qualitatively.

I recommend the authors to give more details on the protocol and Yoga sessions conducted. The frequency, the yoga poses adapted or not adapted to the needs of the participants.

Since, YOGA can be individualized as movement intervention, I would like to see the how the increased awareness and knowledge on MS could contribute to the further individualization to YOGA sessions.

It is possible to discuss the role of the breathing exercises more on the positive feelings and wellbeing. Moreover, if the combination of Yoga poses and breathing augmented the effect, it would be good to have some science based ideas about this link.

Please describe if there is any specific information on the degree or level of participant's MS even the average age.

The words "sense of working hard" and "sense of achievement" could be mentioned in the results and discussion to strengthen the link between the conclusion and the results.

Data is available due to the condition for the ethical approval.

Reviewer #2: I believe that this is an interesting study and it has a merit for this journal. However, this manuscripts needs substantial development and editing. Especially, a strong introduction, discussion and conclusion parts needed with up to date literature review and (2020 and above) and practical suggestions for the future researchers. A strong theoretical background needed in manuscript. In addition, reliability and validity of the data should be reported. In summary, the study needs more clarification in terms of theory, background and reliability and validity of the data. Also, each individual characteristics should be defined in detail such as gender, race, education and culture. Best Regards.

6. PLOS authors have the option to publish the peer review history of their article (what does this mean?). If published, this will include your full peer review and any attached files.

Reviewer #1: **Yes: **Ertan TUFEKCIOGLU

Reviewer #2: **Yes: **Ferman Konukman

---

## [Author Response · Author response to Decision Letter 0]

29 Apr 2023

Dear reviewers,

Very many thanks for your helpful comments on the manuscript. We have addressed all areas and feel this is a much-improved paper. The responses to each reviewer point are in the table below.

Information about informed consent has been added to lines 129-133. The data availability statement is located on lines 548-552. 

The response to reviewers has been uploaded as a word document including a table with each comment and response. 

Kind regards

Dr Jenni Naisby

---

## [Decision Letter · Decision Letter 1]

26 Jun 2023

Yoga ve Multipl Skleroz: Fiziksel Aktiviteye Bağlılığı Sürdürme 

PONE-D-22-34381R1

Sevgili Doktor Jenni Naisby,

Makalenizin bilimsel olarak yayına uygun olduğuna karar verildiğini ve tüm üstün teknik gereklilikleri karşıladığında yayın için resmi olarak kabul edileceğini size bildirmekten memnuniyet duyuyoruz.

Bir hafta içinde gerekli değişiklikleri içeren bir e-posta alacaksınız. Bunlar ele alındığında, resmi bir kabul mektubu alacaksınız ve makalenizin yayınlanması planlanacak.

Resmi kabulden kısa bir süre sonra bir ödeme faturası gelir. Verimli bir süreç sağlamak için, lütfen http://www.editorialmanager.com/pone/ adresinden Yayın Yöneticisi'nde oturum açın, sayfanın üst kısmındaki 'Bilgilerimi Güncelle' bağlantısını tıklayın ve kullanıcı bilgilerinizin güncel olup olmadığını iki kez kontrol edin. bugüne kadar. Faturalandırmayla ilgili herhangi bir sorunuz varsa, lütfen doğrudan Authorbilling@plos.org adresinden Yazar Faturalandırma departmanımızla iletişime geçin.

Kurumunuzun veya kuruluşlarınızın bir basın ofisi varsa, etkisini en üst düzeye çıkarmak için lütfen yaklaşan makaleniz hakkında onları bilgilendirin. Basın materyallerini hazırlayacaklarsa, lütfen resmi kabulü aldıktan sonra en geç 48 saat içinde basın ekibimizi bilgilendirin. Makaleniz, yayın tarihinde Doğu Saati ile 14:00'e kadar katı basın ambargosu altında kalacaktır. Daha fazla bilgi için lütfen onepress@plos.org ile iletişime geçin.

Saygılarımla,

Bijen Filiz 

Akademik Editör 

PLOS ONE

Ek Editör Yorumları (isteğe bağlı):

Hakemlerin yorumları:

Hakemlerin Sorulara Yanıtları

**Yazara Yorumlar**

1. Yazarlar, önceki bir gözden geçirme turunda dile getirdiğiniz yorumlarınıza yeterince değindiyse ve bu makalenin artık yayın için kabul edilebilir olduğunu düşünüyorsanız, burada "Yazarın Yorumları" bölümünü atlamak için şunu belirtebilirsiniz: "Editöre Gizli" bölümündeki çıkar çatışması bildiriminizi ve "Kabul Et" önerinizi gönderin.

1 Numaralı İnceleyen: Tüm yorumlar ele alındı

2. İnceleyen: Tüm yorumlar ele alındı

2. El yazması teknik olarak sağlam mı ve veriler sonuçları destekliyor mu? 

El yazması, teknik olarak sağlam bir bilimsel araştırma parçasını, sonuçları destekleyen verilerle açıklamalıdır. Deneyler, uygun kontroller, çoğaltma ve numune boyutları ile titizlikle yürütülmüş olmalıdır. Sonuçlar, sunulan verilere dayanarak uygun şekilde çıkarılmalıdır.

1. İnceleyen: Evet

2. İnceleyen: Evet

3. İstatistiksel analiz uygun ve titiz bir şekilde yapıldı mı?

1. İnceleyen: Evet

2. İnceleyen: Yok

4. Yazarlar, makalelerindeki bulguların altında yatan tüm verileri eksiksiz olarak sağladı mı? 

PLOS Veri politikası, yazarların makalelerinde açıklanan bulguların altında yatan tüm verileri, nadir istisnalar dışında, kısıtlama olmaksızın tamamen erişilebilir hale getirmelerini gerektirir (lütfen el yazması PDF dosyasındaki Veri Kullanılabilirlik Bildirimine bakın). Veriler, makalenin veya destekleyici bilgilerinin bir parçası olarak sağlanmalı veya halka açık bir depoda saklanmalıdır. Örneğin, özet istatistiklere ek olarak, ortalamaların, medyanların ve varyans ölçümlerinin arkasındaki veri noktaları da mevcut olmalıdır. Verilerin herkese açık olarak paylaşılmasına ilişkin kısıtlamalar varsa - örneğin, katılımcı gizliliği veya üçüncü şahıslardan alınan verilerin kullanımı - bunlar belirtilmelidir.

1. İnceleyen: Hayır

2. İnceleyen: Evet

5. Makale anlaşılır bir şekilde sunuldu ve standart İngilizce ile yazılmış mı? 

PLOS ONE kabul edilen makaleleri kopyalayıp düzenlemez, bu nedenle gönderilen makalelerdeki dil açık, doğru ve net olmalıdır. Herhangi bir tipografik veya gramer hatası revizyonda düzeltilmelidir, bu nedenle lütfen belirli hataları burada not edin.

1. İnceleyen: Evet

2. İnceleyen: Evet

6. Yazarın Yorumlarını Gözden Geçirin 

Lütfen yukarıdaki sorulara verdiğiniz cevapları açıklamak için verilen alanı kullanınız. Yazar için ikili yayın, araştırma etiği veya yayın etiği ile ilgili endişeler dahil olmak üzere ek yorumlar da ekleyebilirsiniz. (Lütfen yorumunuzu 20.000 karakteri aşarsa ek olarak yükleyin)

1 Numaralı İnceleyici: İncelemeci yorumlarının çoğuna değindiğiniz için teşekkür ederiz. Bir alandaki ilk çalışmalardan biri olmanın sınırlamaları olduğunu anlıyorum. Ancak, çalışma sınırları kabul edilebilir. 

Yazarları bu yüksek kaliteli çalışma için tebrik etmek istiyorum. 

Saygılarımızla

2. Hakem: Bu ilginç çalışma için yazarlara teşekkür etmek istiyorum. Bu yazının mevcut durumunun yayın için kabul edilebilir olduğuna inanıyorum. Saygılarımla.

7. PLOS yazarları, makalelerinin akran değerlendirmesi geçmişini yayınlama seçeneğine sahiptir ( bu ne anlama geliyor? ). Yayınlanırsa, bu, tam akran değerlendirmenizi ve ekli dosyaları içerecektir. 

"Hayır"ı seçerseniz, kimliğiniz anonim kalır ancak incelemeniz yine de herkese açık olabilir. 

**Bu akran değerlendirmesi için kimliğinizin herkese açık olmasını istiyor musunuz? **Rızanın geri alınması da dahil olmak üzere bu seçim hakkında bilgi için lütfen Gizlilik Politikamıza bakın .

Yorumcu #1:  **Evet: ** Ertan Tüfekçioğlu

Yorumcu #2:  **Evet: ** Ferman Konukman

---

## [Editor Report · Acceptance letter]

11 Jul 2023

PONE-D-22-34381R1 

Yoga and Multiple Sclerosis: Maintaining Engagement in Physical Activity 

Dear Dr. Naisby:

I'm pleased to inform you that your manuscript has been deemed suitable for publication in PLOS ONE. Congratulations! Your manuscript is now with our production department. 

Kind regards, 

on behalf of

Dr. Bijen Filiz 

Academic Editor

PLOS ONE